# THE FEATURE-SPACE ALIGNMENT HYPOTHESIS FOR NEURAL NETWORK SPARSITY

**Linghao Kong**[1], **Micah Adler**[1], **& Nir Shavit**[1,2]
[1]MIT   [2]Red Hat AI
{linghao, micah432, shanir}@mit.edu

## ABSTRACT

Why does something as simple as magnitude pruning succeed even when most weights are removed, and why does it eventually fail? We study this in a controlled setting where the network's feature-space weights are explicit. We find that accuracy collapse under standard pruning coincides with divergence of this feature-space representation from its dense counterpart. We then introduce an optimization oracle that selects sparse weight matrices independent of the original that explicitly preserve the latent feature structure. Under identical retraining budgets, the oracle recovers performance at sparsities where standard pruning degrades across both logic and vision tasks. This suggests that sparsification limits can arise from misalignment between weight-space and feature-level structure, and points to feature-aware criteria as a path toward improved pruning methods.

## 1 INTRODUCTION

Weight sparsity has long been a goal in neural network research (LeCun et al., 1989; Hassibi & Stork, 1992; Han et al., 2015; Frankle & Carbin, 2018; Frantar & Alistarh, 2023; Sun et al., 2023) with increasing relevance (Dubey et al., 2024; Team et al., 2025; Liu et al., 2025). Most pruning methods operate in weight space, pruning parameters by magnitude or local sensitivity (LeCun et al., 1989; Hassibi & Stork, 1992; Frankle & Carbin, 2018; Frantar & Alistarh, 2023). This implicitly assumes that features, circuits, and decision-relevant computations align with individual parameters.

However, growing evidence suggests that features are often distributed and entangled (Goh et al., 2021; Olah et al., 2020; Arora et al., 2018; Mu & Andreas, 2020; Jermyn et al., 2022; Gurnee et al., 2024; Dreyer et al., 2024; Elhage et al., 2022; Adler & Shavit, 2024), making weight-based heuristics misaligned with feature preservation. As a result, pruning even nominally unimportant parameters can distort computation (Sawmya et al., 2025), potentially contributing to brittleness at higher sparsities (Agarwalla et al., 2024; Lee et al., 2025). We term this mismatch between parameter-level pruning objectives and feature-level structure the *feature-space alignment hypothesis*.

To study misalignment between weights and features, we introduce a controlled setup that separates sparsity in the raw weights from behavior in an explicit feature-space representation, mirroring a standard layer while exposing typically hidden feature structure. We find pruning succeeds when it preserves these latent features and fails when it destroys them. Across both a Boolean logic task and a vision task, we show that for the same sparsity, an optimization oracle can select sparse weights that preserve the effective features and recover performance where standard methods fail. Finally, we find that the topology largely determines achievable performance: while oracle masks require weight adaptation, after fine-tuning they outperform standard pruning, suggesting that observed sparsity limits can arise from basis misalignment rather than the capacity of the sparse network.

## 2 METHODOLOGY

We model a single layer of a network using a canonical setup, with an added embedding layer as established in Adler et al. (2025). Formally, given inputs $\mathbf{x} \in \mathbb{R}^{d_{in}}$, we first apply a learnable linear projection $\mathbf{C}_0 \in \mathbb{R}^{d \times d_{in}}$ to obtain a transformed representation $\mathbf{x}' = \mathbf{C}_0 \mathbf{x}$, simulating prior layers in a deep network. The network then computes output logits $\mathbf{z}$ via $\mathbf{z} = \mathbf{W}_2 \cdot \text{ReLU}(\mathbf{W}_1 \mathbf{x}' + \mathbf{b}_1) + \mathbf{b}_2$,

where $\mathbf{W}_1 \in \mathbb{R}^{h \times d}$ and $\mathbf{W}_2 \in \mathbb{R}^{C \times h}$ are the weight matrices for the hidden and output layers, respectively, and $\mathbf{b}_1, \mathbf{b}_2$ are biases. To isolate the effect of feature-weight alignment, we avoid introducing dimensionality changes: we use a square linear embedding $\mathbf{C}_0$ whose output dimensionality matches its input dimensionality and set the width of $\mathbf{W}_1$ to the same value (section A.2).

$\mathbf{C}_0$ is central to our design. Adler et al. (2025) showed that the network is implicitly learning $\mathbf{W}_1 = \mathbf{C}_1\mathbf{C}_0^{-1}$, where $\mathbf{C}_1$ represents the feature space computation: $\mathbf{W}_1$ "disentangles" the relevant features from $\mathbf{x}'$ by inverting $\mathbf{C}_0$, then performs the computation in feature-space. This allows us to test how pruning $\mathbf{W}_1$ affects the underlying feature space: $\mathbf{C}_1 = \mathbf{W}_1\mathbf{C}_0 \in \mathbb{R}^{h \times d_{in}}$. Thus, $\mathbf{C}_1$ will be the primary object of analysis for interpreting internal representations under pruning, distinguishing functional structure in $\mathbf{C}_1$ from sparsity in $\mathbf{W}_1$ (fig. 3).

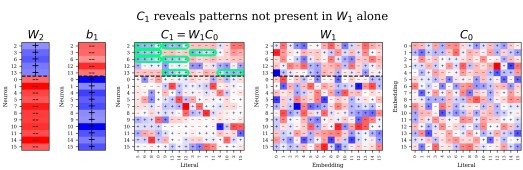

Figure 1: Visualizing weights in $\mathbf{C}_1$ reveals codes that are not present in $\mathbf{W}_1$. Sets of weights circled in green indicate 4P-positive codes, as explained in section 2.

To solidify this intuition, consider fig. 1. The figure shows parameters of a dense model ($\mathbf{C}_0$, $\mathbf{W}_1$, $\mathbf{b}_1$, and $\mathbf{W}_2$), along with $\mathbf{C}_1$, trained to learn a DNF formula of 4-variable clauses. For visualization, both $\mathbf{W}_1$ and $\mathbf{C}_1$ are ordered by clauses along the horizontal axis, and by neurons with positive output weights in $\mathbf{W}_2$ along the vertical axis. We see (highlighted in green) in $\mathbf{C}_1$ the codes that emerge in rows with positive $\mathbf{W}_2$ neurons: 4 consecutive positive weights with a corresponding negative bias. This is a soft Boolean *AND* (Gupta, 1999) of the specific clause $\bigwedge_{j=0}^{3} x_{i+j}$, computed via $\mathrm{ReLU}\left(\sum_{j=0}^{3} w_{i+j}x_{i+j} - b\right)$. We refer to these as "4P-positive" codes. Notably, the codes emerge in feature space, and cannot be seen in the model's weight space where pruning occurs.

We begin in a controlled setting where the features are known and can be tracked during training and pruning, though our technique does not need to know the features. We study learning $k$-uniform read-once monotone DNF formulas (O'Donnell, 2014), a standard benchmark for compositional structure (Valiant, 1984; Daniely et al., 2016; Bronstein et al., 2022). The target function is $f(x) = \bigvee_{j=1}^{m/4} \bigwedge_{\ell \in S_j} x_\ell$, where $\{S_j\}$ form a partition of $\{1, \ldots, m\}$ with $|S_j| = 4$ for all $j$. For a given hidden neuron $i$ and a target clause $S_j$, a code is defined as the joint sign pattern of the feature-space weights $\{(\mathbf{C}_1)_{i,\ell} \mid \ell \in S_j\}$. This determines the literals to which the neuron responds and whether its activation contributes positively or negatively to the output. We focus on the 4P-positive codes computing *AND*. We repeat our experiments on Scikit-learn Digits (Pedregosa et al., 2011).

## 3 RESULTS

We study the structural properties of the learned solution within $\mathbf{C}_1$ across problem instances with 16, 32, 48, and 64 literals to assess how feature formation and pruning effects scale with task complexity. Over training, we observe the consistent emergence of the dominant 4P-positive code (fig. A1), in line with prior work (Adler et al., 2025). While there are other important codes that handle near misses, we focus our subsequent analysis on these codes. We now use this framework to analyze the mechanism underlying performance degradation under sparsity: we track the prevalence of 4P-positive codes in $\mathbf{C}_1$ as $\mathbf{W}_1$ is progressively sparsified via magnitude pruning (Han et al., 2015): pruning of the smallest weights. Here, we keep the same sparsity per neuron.

As sparsity increases, we observe an increase in loss on the training dataset accompanied by a degradation of 4P-positive codes. These two trends closely mirror one another, aligning far stronger than with the degradation of any other pattern (fig. 2). This correlation suggests 4P-positive codes constitute essential feature-level structures supporting computation. We visualize $\mathbf{C}_1$ at increasing sparsity in $\mathbf{W}_1$ (fig. 3). Expectedly, $\mathbf{C}_1$ remains numerically dense even under extreme $\mathbf{W}_1$ sparsity. This density masks a substantial loss of structure: the 4P-positive codes progressively degrade. Thus,

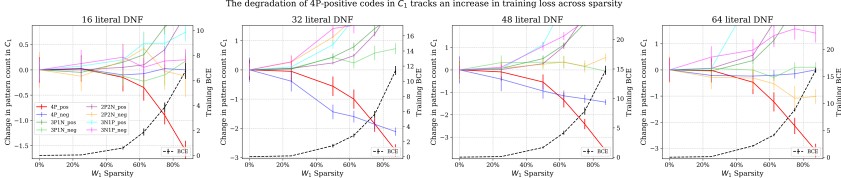

Figure 2: Code and performance degradation under sparsity. As $\mathbf{W}_1$ sparsity via magnitude pruning increases, an increase in loss on the training dataset tracks the loss of 4P-positive codes in $\mathbf{C}_1$. In this and upcoming figures, error bars indicate one standard error, and results collected over ten trials.

the damage from sparsification does not primarily act by making $\mathbf{C}_1$ sparse, but by disrupting its alignment with the dense solution. The corresponding Digits visualization is in fig. A2.

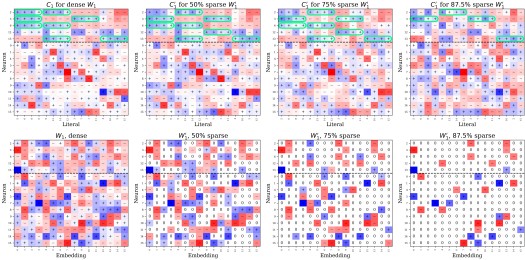

Figure 3: Visualization of $\mathbf{C}_1$ and $\mathbf{W}_1$ under sparsity. Even at high $\mathbf{W}_1'$ sparsity, $\mathbf{C}_1'$ remains fairly dense, highlighting that preserving the structure of $\mathbf{C}_1'$ is relevant. Dense model is same as fig. 1.

We investigate if directly preserving $\mathbf{C}_1$, without knowing what the features are, improves performance. We pose this as a constrained reconstruction problem: given oracle access to $\mathbf{C}_1$ of a trained dense model, we search for a sparse weight matrix $\mathbf{W}_1'$ with at most $\tau$ non-zero entries per neuron whose composition with $\mathbf{C}_0$ best approximates the feature weights, using a per-neuron formulation for tractability. We solve the mixed integer problem $\min_{\mathbf{W}_1', \mathbf{M}} \|\mathbf{W}_1 \mathbf{C}_0 - (\mathbf{M} \odot \mathbf{W}_1')\mathbf{C}_0\|_F^2$, subject to $\sum_{j=1}^d \mathbf{M}_{ij} \leq \tau, \forall i \in \{1, \ldots, h\}, \mathbf{M}_{ij} \in \{0, 1\}, \mathbf{W}_1' \in \mathbb{R}^{h \times d}$, where $\mathbf{M}$ is a binary mask. This yields a sparse approximation that is optimal with respect to preserving $\mathbf{C}_1$ and separates topology ($\mathbf{M}$) from weight values ($\mathbf{W}_1'$). The MIP solution departs further from the dense model in parameter space than magnitude pruning, while remaining closer in feature space (fig. A3).

We compare the performance of our optimization oracle against three standard baselines: magnitude pruning (MP) (Han et al., 2015), Optimal Brain Damage (OBD) (LeCun et al., 1989), and Optimal Brain Surgeon (OBS) (Hassibi & Stork, 1992). The same sparsity level per neuron in $\mathbf{W}_1$ was enforced for all techniques. MP removes weights in $\mathbf{W}_1$ with the smallest absolute values. OBD refines MP by incorporating a diagonal approximation to the Hessian, pruning weights estimated to cause the smallest increase in loss. OBS further extends this approach by accounting for interactions between weights and applies a one-shot compensation to the remaining weights. Full details in section A.3. In contrast, **M**ixed **I**nteger **P**rogramming (MIP) directly optimizes for a sparse weight matrix $\mathbf{W}_1'$ by minimizing distortion in the effective feature space $(\mathbf{M} \odot \mathbf{W}_1')\mathbf{C}_0$ relative to $\mathbf{W}_1 \mathbf{C}_0$. Finally, the **F**eature **Lo**ss Hybrid (FLO) method uses the sparse mask $\mathbf{M}$ identified by MIP while keeping the original weights, followed by an OBS-style weight update, decoupling topology selection from weight refinement. Each technique induces a distinct sparse mask on $\mathbf{W}_1$ (fig. A4)

We first compare the performance of MP, OBD, OBS, and MIP after pruning without any retraining. We observe that MIP and OBS achieve broadly comparable performance, with both substantially outperforming MP and OBD (fig. 4). This parity suggests that preserving feature-space structure (as in MIP) and compensating for pruning-induced error via second-order updates (as in OBS) address complementary aspects of sparsification. Motivated by these complementary strengths, we combine the MIP sparse topology with the OBS weight update in FLO. FLO outperforms all other methods, indicating that accurate identification of a feature-aligned mask and effective post-pruning weight compensation jointly contribute to improved performance. By explicitly reconstructing $\mathbf{C}_1$,

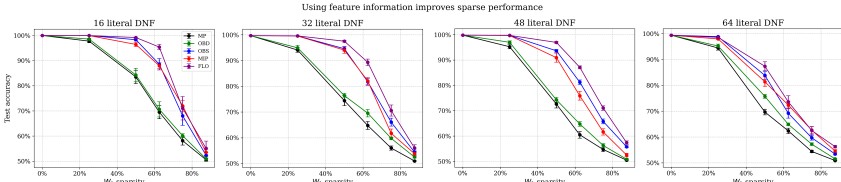

Figure 4: Feature-space information improves sparse performance. OBS and MIP have comparable test accuracy performance after pruning of $\mathbf{W}_1$ with no fine-tuning, while FLO outperforms both.

MIP identifies a sparse support that preserves the network's function more effectively than saliency heuristics derived from $\mathbf{W}_1$. These results are consistent for structured sparsity (section A.9).

To isolate topology selection from weight optimization, we evaluate each method using only its sparse mask $\mathbf{M}$, keeping all weights at their dense values (affecting only OBS and MIP). MIP initially underperforms MP, indicating that oracle-optimized masks do not transfer directly. However, with the same fine-tuning budget and training to saturation, the MIP-derived topology consistently achieves the best final performance (fig. 5), revealing a distinction between initialization quality and representational capacity. The oracle topology provides a superior substrate for restoring feature-space structure during fine-tuning, whereas standard heuristics discard low-saliency connections that remain feature-critical, leading to lower performance ceilings. These heuristics converge to similar final accuracies, suggesting they induce comparable topologies that impose a shared recovery limit.

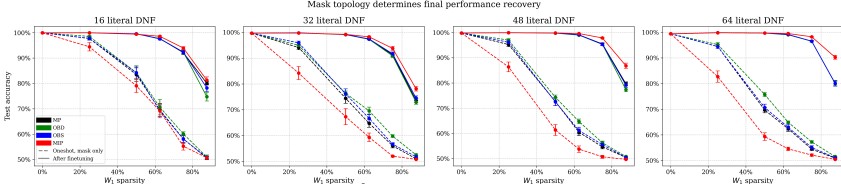

Figure 5: Mask topology determines performance recovery under sparsification. With equal fine-tuning budget, the MIP mask allows the model to recover more test accuracy.

We find that fine-tuning largely preserves existing code structure, indicating that recoverable performance is primarily determined by whether pruning retains a sufficiently rich feature substrate (fig. A6). Indeed, MIP preserves more 4P-positive codes than MP, OBD, or OBS across sparsities (fig. A7), despite not explicitly targeting them. We stress that this relationship is correlational: these codes serve as an interpretable proxy for the stability of functionally meaningful structure in $\mathbf{C}_1$.

Digits involves continuous inputs and geometric feature extraction. Here, feature-aware methods also achieve the highest sparse performance, with FLO outperforming all other techniques. OBS outperforms MIP, reflecting the increased importance of weight adaptation in continuous tasks. Nevertheless, FLO's strong performance indicates that accurate topological selection remains essential, and that combining feature-aligned masks with effective weight updates is beneficial in more realistic settings. MP, OBD, OBS, and MIP again induce qualitatively different sparsity masks (fig. A5).

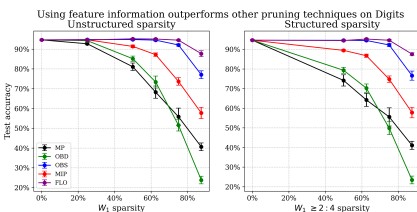

Figure 6: Feature-space information improves Digits performance.

Overall, our results suggest that pruning failures can arise from misalignment between weight-space heuristics and the feature-space structure where computation occurs.

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

# A   APPENDIX

## A.1   DATASET GENERATION DETAILS

We construct a synthetic Boolean dataset based on monotone disjunctive normal form (DNF) formulas, designed to allow controlled analysis of feature-level structure under sparsification.

**DNF Structure.**   For a Boolean formula with $m$ input literals, where $m \in \{16, 32, 48, 64\}$ in all experiments, we partition the literals into $m/4$ disjoint clauses. Each clause consists of exactly four literals, and no literal appears in more than one clause. The resulting formula is therefore a *read-once monotone DNF* of the form

$$f(x) = \bigvee_{j=1}^{m/4} \bigwedge_{\ell \in S_j} x_\ell,$$

where $\{S_j\}$ form a partition of $\{1, \ldots, m\}$ with $|S_j| = 4$ for all $j$.

**Input Sampling.**   Each input sample $x \in \mathbb{R}^m$ is initialized to zero. For each sample, we uniformly draw an integer $k$ between $m/4$ and $1.5 \cdot (m/4)$, and randomly select $k$ distinct literals to activate. Activated literals are set to a value of $1.5$ rather than $1$, which we found improves training stability, while all remaining literals remain at zero.

**Dataset Size.**   The total number of training samples is set to $50{,}000 \cdot (m/16)$, and the total number of test samples is set to $10{,}000 \cdot (m/16)$, ensuring that dataset size scales linearly with input dimensionality.

**Label Assignment and Balancing.**   Each generated sample is evaluated under the target DNF. Samples that satisfy at least one clause are labeled positive, while all others are labeled negative. To maintain a balanced dataset, we track the number of positive and negative samples during construction and discard any sample that would cause either class to exceed half of the total dataset size. This procedure yields approximately balanced class proportions without explicit resampling.

## A.2 TRAINING DETAILS

All models are trained using the same optimization and batching procedure across datasets and sparsity settings in PyTorch (Paszke et al., 2019).

**Optimization Setup.** Training is performed using the Adam optimizer with default momentum parameters. All datasets are trained with a fixed batch size of 256.

**Base Training.** All models are trained for 25 epochs with a learning rate of $10^{-3}$. This training procedure is applied uniformly across all Boolean and vision datasets considered in the paper.

**Fine-Tuning.** When fine-tuning is applied after pruning, we continue training the resulting sparse model for an additional 15 epochs using a reduced learning rate of $10^{-4}$. Fine-tuning also uses the Adam optimizer with default momentum parameters, and the same batch size as the base training phase.

No additional learning rate schedules, regularization terms, or early stopping criteria are used.

**Code Emergence.** Across all experimental conditions, the 4P-positive codes arise in tandem with a decrease in training loss (fig. A1).

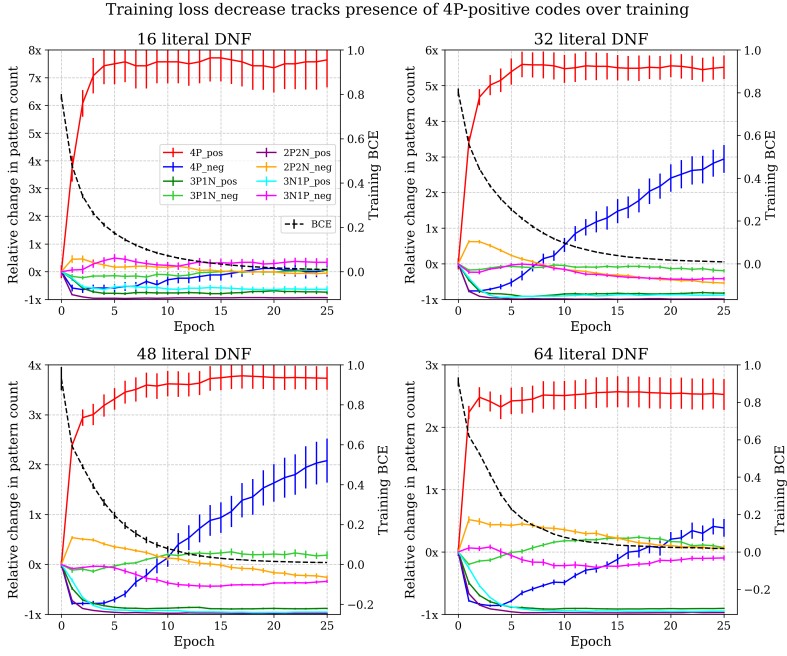

Figure A1: Prevalence of different codes during training. Network performance tracks the prevalence of 4P-positive codes during training. This thus motivates our tracking of such codes over the course of sparsification. 4N codes not shown since their effect after ReLU is zero. Error bars indicate one standard error of the mean. Results collected over ten trials.

### A.3 Sparsification Technique Details

We briefly describe the saliency criteria used by the second-order pruning baselines considered in this work, following the classical formulations of Optimal Brain Damage (OBD) (LeCun et al., 1989) and Optimal Brain Surgeon (OBS) (Hassibi & Stork, 1992). Both methods estimate the increase in loss induced by removing individual weights using a local quadratic approximation.

**Second-Order Loss Approximation.** Let $\mathcal{L}(\mathbf{w})$ denote the training loss as a function of the model parameters $\mathbf{w}$. Around a local minimum $\mathbf{w}$, the loss can be approximated by a second-order Taylor expansion:

$$\mathcal{L}(\mathbf{w} + \Delta\mathbf{w}) \approx \mathcal{L}(\mathbf{w}) + \nabla\mathcal{L}(\mathbf{w})^\top \Delta\mathbf{w} + \tfrac{1}{2}\Delta\mathbf{w}^\top \mathbf{H}\Delta\mathbf{w},$$

where $\mathbf{H}$ is the Hessian of $\mathcal{L}$ with respect to $\mathbf{w}$. At a local optimum, the gradient term vanishes, leaving only the quadratic contribution. In our experiments, we use $10\%$ of the samples in the training dataset to calculate the Hessian the for the boolean task and the entire training dataset for the Digits task.

**Optimal Brain Damage (OBD).** OBD assumes that removing a single weight $w_i$ does not induce compensatory changes in the remaining parameters. Under this assumption, pruning $w_i$ corresponds to the perturbation $\Delta\mathbf{w} = -w_i\mathbf{e}_i$, where $\mathbf{e}_i$ is the $i$-th standard basis vector. Substituting into the quadratic approximation yields

$$\Delta\mathcal{L}_i \approx \tfrac{1}{2}\mathbf{H}_{ii}w_i^2.$$

This quantity defines the *saliency* of weight $w_i$. OBD prunes the weights in $\mathbf{W}_1$ with the smallest saliency values, resulting in a curvature-aware refinement of magnitude pruning that accounts for local sensitivity of the loss.

**Optimal Brain Surgeon (OBS).** OBS relaxes the independence assumption of OBD by allowing the remaining weights to adjust in order to minimize the loss increase caused by pruning. The method considers the constrained optimization problem of removing a weight $w_i$ while optimally updating all other parameters to compensate.

Formally, pruning $w_i$ imposes the constraint $\Delta w_i = -w_i$. Minimizing the quadratic loss approximation subject to this constraint yields the optimal parameter update

$$\Delta\mathbf{w} = -\frac{w_i}{(\mathbf{H}^{-1})_{ii}}\mathbf{H}^{-1}_{\cdot i},$$

and an associated increase in loss given by

$$\Delta\mathcal{L}_i = \tfrac{w_i^2}{2(\mathbf{H}^{-1})_{ii}}.$$

This expression defines the OBS saliency criterion, which explicitly accounts for interactions between parameters through the inverse Hessian. In practice, a damped or approximate inverse Hessian is used for numerical stability.

**OBS Compensation Update.** When pruning a set of weights $S$ simultaneously, OBS applies the compensation update in closed form. Let $\mathbf{w}_S$ denote the subvector of parameters indexed by $S$. The optimal update to the remaining parameters is

$$\Delta\mathbf{w} = -\mathbf{H}^{-1}_{\cdot S}(\mathbf{H}^{-1})^{-1}_{SS}\mathbf{w}_S,$$

which minimizes the second-order loss increase subject to setting all pruned weights to zero. This update is applied once after mask selection, yielding a sparsified model that preserves performance more effectively than uncorrected pruning. Note that this update is independent of the choice of mask, which allows us to use an OBS-style weight compensation update with the MIP mask as our FLO condition.

**Mixed Integer Programming.** We implement this formulation in Python using CVXPY (Diamond & Boyd, 2016) with SCIP (Hojny et al., 2025) as the underlying solver.

## A.4 Visualizations of $\mathbf{C}_1$ with Increasing $\mathbf{W}_1$ Sparsity

We repeat the same sparsification setup for the Digits task. The Scikit-learn Digits dataset (Pedregosa et al., 2011) consists of $8 \times 8$ grayscale images of handwritten digits (0–9), providing a simple but non-synthetic benchmark that captures real-world visual structure while remaining computationally lightweight. As shown in fig. A2, similarly to the DNF task, increasing sparsity rapidly destroys visible structure in $\mathbf{W}_1$, yet the induced feature-space matrix $\mathbf{C}_1$ remains reasonably close to the dense solution up to moderate sparsity levels, before degrading sharply at higher sparsities. This is reflected by a drop in accuracy (fig. 6). This behavior may help explain why magnitude pruning works well at moderate sparsities. It also motivates, as we will see in the next section, methods that explicitly preserve $\mathbf{C}_1$ rather than relying on implicit alignment.

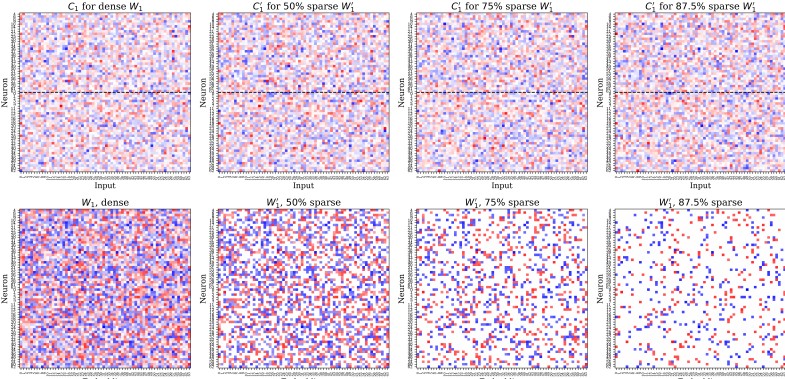

Figure A2: Another example of the effect of magnitude pruning on weight and feature-space representations, this time on *Digits*, an MNIST (LeCun et al., 2002) style task. The scaled $L_2$ norms between the dense and sparse $\mathbf{C}_1$ matrices is 0.38, 0.65, and 0.81 for 50%, 75%, and 87.5% $\mathbf{W}_1$ sparsity, respectively.

## A.5 DIRECTLY MINIMIZING DIVERGENCE FROM $\mathbf{C}_1$

Figure A3: Divergence of $\mathbf{W}_1$ and $\mathbf{C}_1$. Despite MIP inducing a greater relative divergence in $\mathbf{W}_1$ from the dense model compared to magnitude pruning, it induces a lesser relative divergence in $\mathbf{C}_1$. Error bars indicate one standard error of the mean. Results collected over ten trials.

## A.6   TOPOLOGY DIFFERENCES IN PRUNING TECHNIQUES

MP, OBD, OBS, and MIP induces a distinct sparse mask on $\mathbf{W}_1$, reflecting different notions of parameter importance (fig. A4)

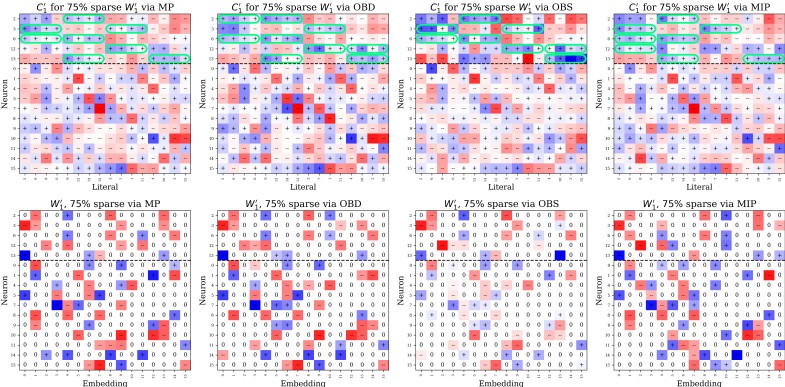

Figure A4: MP, OBD, OBS, and MIP optimize their masks for sparsity differently. Dense model is same as fig. 1.

We can also observe each of these techniques inducing a qualitatively different sparsity masks on digit classification tasks, again reflecting systematic differences in the features preserved by each method (fig. A5).

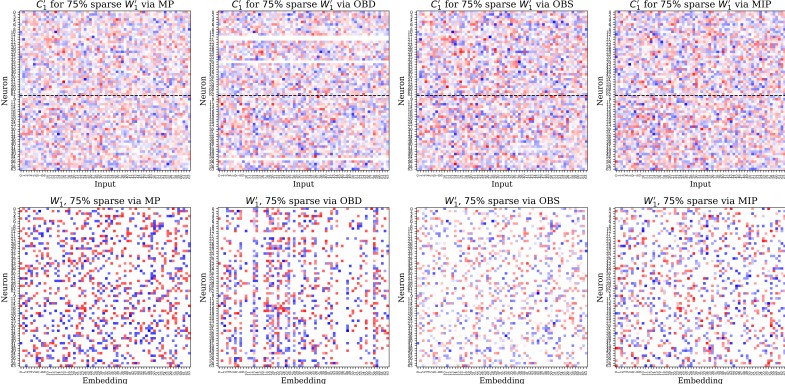

Figure A5: Differences across the pruning techniques for Digits.

## A.7 FINETUNING PRIMARILY REFINES RATHER THAN CREATES

We observe that fine-tuning leaves the overall code structure largely intact. Codes that are present immediately after pruning tend to persist throughout fine-tuning, while codes that are absent in the post-pruning effective mapping are rarely created during fine-tuning, indicating that optimization primarily rebalances and amplifies surviving structure rather than reliably reconstructing missing features. These observations support the view that final recoverable performance is largely determined by whether the pruning step preserves a sufficiently rich code substrate (fig. A6).

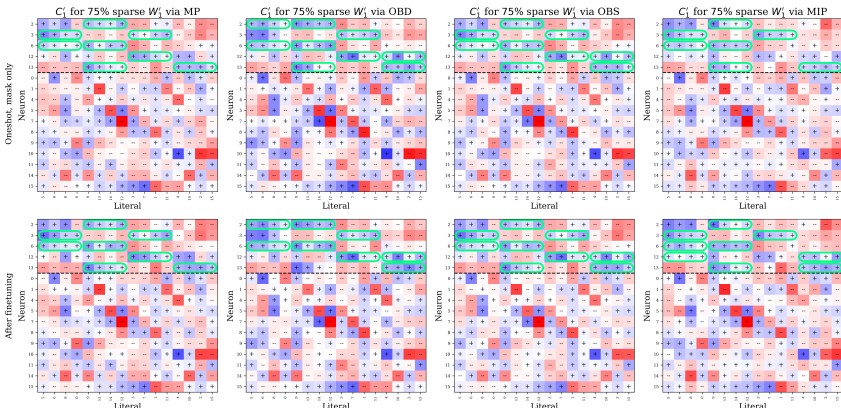

Figure A6: Finetuning largely preserves the same set of codes. Across MP, OBD, OBS, and MIP, fine-tuning Dense model is same as fig. 1.

## A.8 MIP BETTER MAINTAINS 4P-POSITIVE CODES

We analyze the prevalence of 4P-positive codes under increasing sparsity. As shown in fig. A7, MIP consistently preserves a larger number of 4P-positive codes compared to MP, OBD, and OBS. Notably, this behavior emerges despite the fact that the MIP objective does not explicitly target these codes, but instead optimizes reconstruction of the full feature-space $C_1$. The resulting retention of 4P-positive codes therefore arises as a byproduct of minimizing feature-space reconstruction error, rather than from direct supervision of any specific code pattern.

We emphasize that this correlation is not causal. Rather, these codes serve as an interpretable proxy for the stability of functionally meaningful structure in $C_1$. Their preferential preservation under MIP is indicative of a broader robustness of the learned feature organization, helping to explain why MIP-selected masks support higher recoverable performance after fine-tuning.

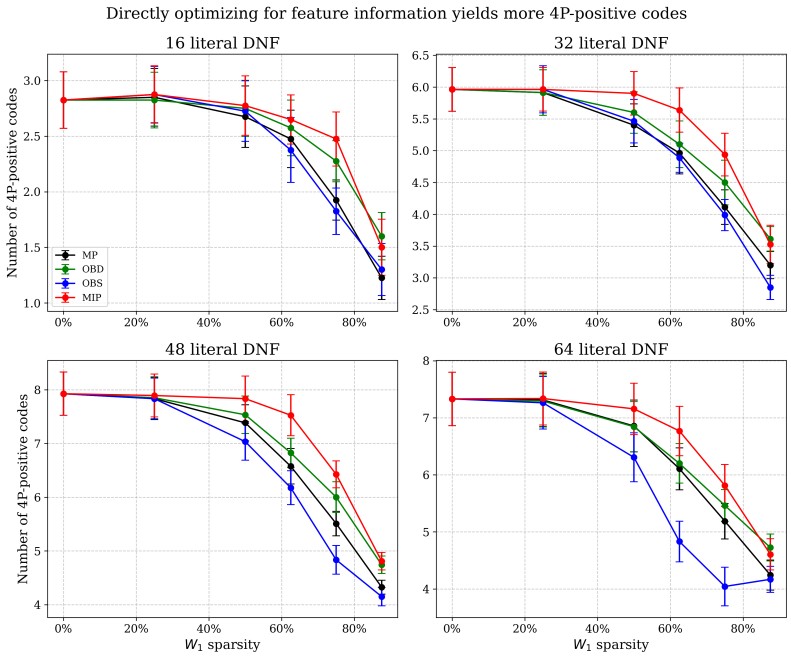

Figure A7: 4P-positive codes are better maintained by MIP. Error bars indicate one standard error of the mean. Results collected over ten trials.

## A.9   STRUCTURED SPARSITY

We extend our MIP formulation to enforce N:M structured sparsity, a constraint widely used for hardware acceleration where $N$ weights are zero in every block of $M$ contiguous weights (Pool et al., 2021). Specifically, we study "$\geq 2 : 4$" sparsity, where at least $2$ of every block of $4$ contiguous weights are zero. Even under these rigid structural constraints, FLO still achieves the highest accuracy, followed by MIP and OBS, with OBD and MP trailing behind. This suggests that the principle of "effective alignment" is robust to the specific granularity of the pruning mask, and that hardware-efficient masks can still support high-fidelity effective weights if selected via the correct objective.

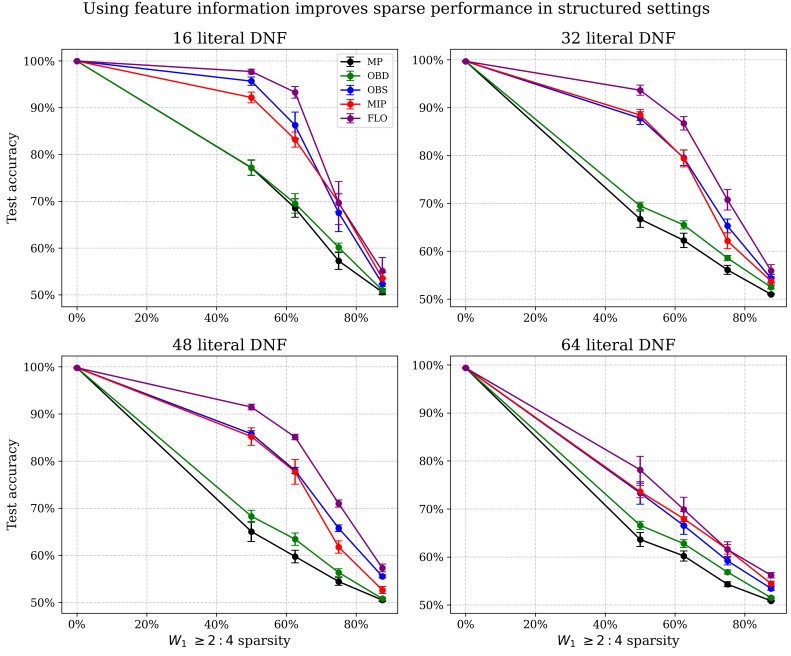

Figure A8: Feature-space information improves sparse performance even in structured settings. As observed previously, OBS and MIP have comparable test accuracy performance after pruning of $\mathbf{W}_1$ with no fine-tuning, while FLO outperforms both. Error bars indicate one standard error of the mean. Results collected over ten trials.

## A.10  Related Work

Despite extensive work on both pruning and mechanistic interpretability, the two have largely progressed separately: pruning typically acts in parameter space, while interpretability emphasizes that computation is expressed in distributed feature bases not aligned with individual weights or neurons. Consequently, pruning rarely accounts for learned changes of basis, and interpretability tools are seldom used to guide compression. We connect these perspectives by arguing that misalignment between the parameter basis and the feature-aligned basis of computation can be a primary bottleneck for sparse performance; the feature-aligned basis is therefore central for understanding when sparsification succeeds or fails.

### A.10.1  Sparsity

Pruning typically identifies unimportant parameters via heuristics or local sensitivity measures. Magnitude pruning (Han et al., 2015) and second-order methods such as Optimal Brain Damage and Optimal Brain Surgeon (LeCun et al., 1989; Hassibi & Stork, 1992) show that many weights can be removed with limited immediate loss. Combinatorial approaches have also been explored, but prior work optimizes OBS-style loss objectives over weights rather than sparsifying with respect to feature-space representations (Yu et al., 2022). At larger scales, one-shot methods demonstrate substantial practical sparsity (Frantar & Alistarh, 2023; Sun et al., 2023), yet pruning outcomes can be brittle and sensitive to training details (Gale et al., 2019; Liu et al., 2018). We provide a mechanistic account of this brittleness: under learned changes of basis, parameter-space saliency can be systematically misaligned with the feature-space directions implementing the computation.

### A.10.2  Interpretability

Mechanistic interpretability argues that networks can represent more features than neurons by storing them in distributed/superposed form (Elhage et al., 2022; Adler & Shavit, 2024). Dictionary-learning and sparse autoencoder methods recover sparse feature directions from dense activations, revealing feature vocabularies far larger than the neuron basis (Bricken et al., 2023; Templeton, 2024; Huben et al., 2023) and echoing classical sparse coding, where sparsity is defined relative to a learned dictionary (Olshausen & Field, 2004). However, these tools are mainly used to measure feature extractability or representation quality (Alain & Bengio, 2016), not to guide compression. Only recently has work examined the interaction between distributed representations and sparsity in weights (Sawmya et al., 2025) and activations (Kong et al., 2025). We build on this line by arguing that sparsity should be defined/enforced in the feature-aligned space of computation, and by showing how misalignment between feature- and parameter-space induces apparent sparsification limits.

## A.11  Limitations and Future Work

Our analysis assumes oracle access to $C_0$ and uses a mixed-integer program, which is NP-hard (Karp, 2009): it is therefore not a practical pruning method for large models. Instead, it quantifies the gap between weight-basis heuristics and feature-aware selection, isolating basis misalignment as a sparsification failure mode.

To scale these insights, one direction is to approximate the feature-aligned basis with sparse autoencoders (SAEs) as suggested in Adler et al. (2025). In our setting, $C_0$ is known so $C_1$ is computable; in realistic networks the analogue of $C_0$ is unknown, but SAEs can recover sparse latent directions from activations. Training an SAE on pre-activations could thus learn a sparse feature dictionary (or approximate inverse embedding), enabling pruning to select masks that preserve feature-level structure without privileged access. Testing such pipelines on modern transformers and downstream tasks would directly assess whether feature-space sparsification can narrow the gap to code-preserving, oracle-style compression.

