# OpenReview forum: "The Feature-Space Alignment Hypothesis for Neural Network Sparsity"
_ICLR.cc/2026/Workshop/Sci4DL — Sci4DL 2026_

### Official Review · Reviewer_WM8o · 2026-02-23

**Fit:** 3
**Significance:** 2
**Confidence:** 2

**Summary:**

The paper explores strategies for pruning weight matrices and presents an approach that is more robust than magnitude pruning.

**Strengths:**

The paper presents rigorous research into the important issue of sparsity.

**Suggestions:**

The writing style is particularly dense even for this style of workshop. This reviewer comes from a signal processing background and often, dense ideas are presented. As this is a newly evolving field, it makes sense for papers to reduce barriers to interpretability from diverse disciplinary backgrounds. One example is "DNF" which isn't expanded until well into the appendix.

There are also some errors and omissions in notation. In particular check the paragraphs before and after fig 1. Also checl lines 386/7 which makes an erroneous statement about the size of each clause. It can't be both 1/4 of the total AND 4 in all cases.

I would also like to see a defined Conclusions section and for it to be more substantial than what seems to be the current conclusions. only line 215 seems to qualify. Given that the workshop is Science, it seems appropriate to follow scientific writing principles.

also, it's a bit unusual to refer to fig 3 before referring to fig 1. This rather suggests renumbering them.

---

### Official Review · Reviewer_4r45 · 2026-02-24

**Fit:** 3
**Significance:** 2
**Confidence:** 3

**Summary:**

The paper explores, in a simple setting, why performance degrades when pruning a neural network. They show that this sparsification limit arises when feature and weight spaces are misaligned. They conducted experiments with different pruning strategies on 4P-positive codes and Scikit-learn Digits datasets.

**Strengths:**

- Clear problem and strong motivation: Identify why and when pruning worsens dense model performance.

**Suggestions:**

- Limited novelty: Exploring how pruning affects training dynamics is largely explored in the literature; the related work section does not cite any papers. Authors also mention leveraging sparse autoencoders to drive model pruning, so papers that exploit interpretability to guide pruning [1,2] or explore how interpretability affects pruning [3] should be mentioned.
- Method: How to explain the phenomenon where, at some sparsity ratios, pruning can improve performance compared to the dense model? Authors show most results when the model is pruned but not fine-tuned; however, fine-tuning for a few epochs is a central step in most pruning methods.
- Experimental setting: Authors should conduct more experiments on more complex datasets and networks.
- Minors: Fig.6 is not mentioned in the main paper, row 076 DNF is not defined

[1] Yeom, Seul-Ki, Philipp Seegerer, Sebastian Lapuschkin, Simon Wiedemann, Klaus-Robert Müller, and Wojciech Samek. 2021. "Pruning by Explaining: A Novel Criterion for Deep Neural Network Pruning." Pattern Recognition

[2] Abbasi-Asl, Reza, and Bin Yu. 2017. "Interpreting Convolutional Neural Networks through Compression."

[3] Enrico Cassano, Riccardo Renzulli, Andrea Bragagnolo, Marco Grangetto, "When Does Pruning Benefit Vision Representations?", ICIAP 2025

---

### Meta-Review · Area_Chair_2eBp · 2026-03-01

**Recommendation:** Accept

**Metareview:**

I recommend accept based on the reviews.

---

### Decision · Program_Chairs · 2026-03-02

Accept